# Theoretical step approach with 'Three-pillar' device assistance for successful endoscopic transpapillary gallbladder drainage

**Michihiro Yoshida, Itaru Naitoh* * , Kazuki Hayashi, Yasuki Hori* , Akihisa Kato, Kenta Kachi, Go Asano, Hidenori Sahashi, Tadashi Toyohara, Kayoko Kuno, Yusuke Kito, Hiromi Kataoka**

Department of Gastroenterology and Metabolism, Nagoya City University Graduate School of Medical Sciences, Nagoya, Japan

* inaito@med.nagoya-cu.ac.jp

**Data Availability Statement:** All relevant data are within the paper.

**Funding:** The authors received no specific funding for this study.

## Abstract

### Background

Endoscopic transpapillary gallbladder drainage (ETGBD) has been reported as an alternative procedure for acute cholecystitis but remains a challenging procedure.

### Aims

To elucidate the efficacy of a strategic approach for ETGBD that utilizes a four-step classification system and the optional use of 'Three-pillar' assistance with the following devices: cholangioscopy (SpyGlass DS, SG), a flex-type guidewire (Flex-GW), and a 3-Fr microcatheter (3-Fr Micro).

### Methods

A total of 115 patients undergoing ETGBD were studied retrospectively. Characteristics and technical outcomes were compared between conventional ETGBD technique (Classical ETGBD, N = 50) and strategic ETGBD with optional Three-pillar assistance (Strategic ETGBD, N = 65).

### Results

SG-assistance (15/65, 23.1%) was as an excellent troubleshooter in Category 1 (failure to identify the cystic duct [CD] orifice) and Category 2 (failure to advance the GW across the CD takeoff due to unfavorable angle). Flex-GW (19/65, 29.2%) worked for Category 3b (failure of GW access to the GB due to multiple tortuosities). 3-Fr Micro (11/65, 16.9%) was effective for Category 3a (failure of GW access to the GB due to CD obstruction) and Category 4 (failure of drainage stent insertion to the GB). The overall technical success rate was significantly higher for Strategic ETGBD (63/65, 96.9%) compared with Classical ETGBD (36/50, 72.0%) (p = 0.0001).

**Competing interests:** The authors have declared that no competing interests exist.

## Conclusions

Strategic ETGBD, which includes the Three-pillar assistance options of SG in the initial steps, Flex-GW for tortuous CD, and 3-Fr Micro for stenotic CD, achieved a significantly higher success rate than for Classical ETGBD.

## Introduction

Acute cholecystitis (AC) is one of the most common gastroenterological diseases, with an etiology of gallbladder stones in approximately 90% of cases [1, 2]. Cholecystectomy has become the definitive treatment [3, 4]; however, surgery is associated with increased mortality, especially in elderly patients and those with multiple severe comorbidities [4]. When AC becomes severe in such patients who are unfit for surgery, urgent decompressive drainage may be required as a life-saving procedure. Tokyo Guidelines 2018 (TG2018) state that early gallbladder (GB) drainage is essential in patients with severe AC in the presence of multiple organ dysfunction or severe local inflammation [5].

Percutaneous transhepatic GB drainage (PTGBD) is a commonly used traditional technique for primary drainage [6–8]. However, percutaneous drainage cannot be performed in patients with massive ascites, an anatomically inaccessible GB, those at risk of self-removal of the drainage tube, or with a bleeding tendency [9].

Endoscopic retrograde cholangiopancreatography (ERCP) is in common use as a minimally invasive approach to biliary diseases, and endoscopic transpapillary GB drainage (ETGBD) is currently considered the second-line drainage procedure for AC. Although ETGBD has been reported to have a significant success rate [10–15], it has technical limitations and difficulties. From a strategic perspective, there are several chokepoints in the ETGBD procedure: potential challenges in identifying the cystic duct (CD) on cholangiography, in advancing the guidewire (GW) through an unfavorable angle of the CD or CD tortuosities, and in the case of obstruction of the CD due to calculus or malignancy. In our previously presented studies, including several case reports, we have described the use of three novel devices for troubleshooting difficult ETGBD: 1) cholangioscopy [16], 2) a new easily maneuverable flex-type GW [17], and 3) a 3-Fr microcatheter [18, 19]. Each device had particular strengths. Cholangioscopy has a considerable advantage in detecting the CD orifice under direct visualization, the new GW has excellent maneuverability with smooth tracking, and the 3-Fr microcatheter enables selective advancement through deep flexures or in the case of a severely stenotic duct. As a novel technique that enables an emergent solution to be applied to each chokepoint, we introduced use of these three devices. In this concept, cholangioscopic assistance was applied when detecting and passing through the orifice of CD, the new GW was introduced when negotiating a tortuous CD, and the 3-Fr microcatheter was used additionally when advancing through a stenotic CD.

The aim of the study was to evaluate the effectiveness of using three optional devices for assistance as a strategy for improving the success rate of ETGBD.

## Material and methods

### Patients

We retrospectively evaluated 115 patients with AC in whom ETGBD was attempted between 1 April 1 2008 and 31 March 31 2022 at Nagoya City University Graduate School of Medical

Sciences. These patients included poor surgical candidates as well as those with any reason for a physician to avoid or hesitate attempting PTGBD. All patients had been diagnosed with AC, which was classified according to the Japanese diagnostic criteria with severity gradings as defined in TG2018 [20]. The diagnostic criteria and characteristics included clinical symptoms such as right upper quadrant pain and tenderness; indicators of systematic inflammation such as fever, leukocytosis, and/or high CRP level; and compatible imaging findings on ultrasonography (US) or computed tomography (CT), such as thickening of the GB wall or fluid around the GB. The severity of AC was defined using three grades: mild, moderate, and severe.

## Conventional ETGBD procedure

Under moderate sedation using a combination of midazolam and pethidine with the patient in the prone position, ERCP using a side-viewing scope was first started with an ERCP catheter and a 0.025-inch GW. After standard biliary cannulation, endoscopic sphincterotomy was performed in the case of naïve papilla. In the case of comorbid CBD stones, the stones were removed using baskets or retrieval balloon catheters with textbook techniques. Conventional ETGBD procedures were then performed as follows. On cholangiogram, the CBD was imaged with deep contrast injection to visualize the orifice and the route of the CD. The GW was manipulated under fluoroscopic guidance to enable insertion into the GB. This procedure involved the use of standard cannulating catheters, extraction balloons, standard sphincterotomies, and/or swing tip catheters with the ability to swing in the opposite direction. A number of 0.025-inch GWs was chosen according to the endoscopist's preference from among Visi-Glide2 (Olympus, Tokyo, Japan), Radifocus (Termo, Tokyo, Japan), RevoWave (PIOLAX Medical Devices, Yokohama, Japan), and Wrangler (PIOLAX Medical Devices). Once the GW had accessed the GB through the CD, a plastic stent was placed into the GB.

## Theoretical 'Step' approach using 'Three-pillar' assistance

Our institution has been conducting ongoing development of a theoretical strategy for attempting ETGBD, and in 2017 initiated an innovative strategy for difficult ETGBD. Whenever insurmountable obstacles were encountered in an ETGBD attempt using conventional techniques, we classified the cause of failure using a four-step system, as follows. Category 0, failure of biliary cannulation; Category 1, failure to identify the CD orifice; Category 2, failure to advance the GW across the CD takeoff due to unfavorable angle; Category 3a, failure of GW access to the GB due to CD obstruction (tumor, stone impaction, or inflammation); Category 3b, failure of GW access to the GB due to multiple tortuosities; and Category 4, failure of drainage stent insertion to the GB [16] (Figs 1 and 2). We adopted a system that used one or more troubleshooting devices targeted to the category of failure, from among three devices: cholangioscopy, a new easily maneuverable flex-type GW, and a 3-Fr microcatheter. These three techniques were collectively termed 'Three-pillar' assistance. In Category 1 and 2, a single-operator digital cholangioscopic system (SpyGlass DS [SG]; Boston Scientific, Natick, MA) was applied, which enables direct visualization for identification of the orifice of the CD and insertion of the GW across the CD. Two additional techniques were applied in Category 3: a flex-type M-Through GW (Flex-GW; Asahi Intecc Corp., Seto, Japan) for smooth manipulation of the GW during advancement through a tortuous CD, and a 3-Fr microcatheter (3-Fr Micro) (Daimon-ERCP-catheter; Hanaco Medical, Saitama, Japan) which assists GW advancement due to its flexibility. In Category 4, a 3-Fr Micro catheter followed the GW over the flexure to the GB, followed by exchange with a stiffer GW type to loosen the crooked CD and enable GB stenting (Fig 2).

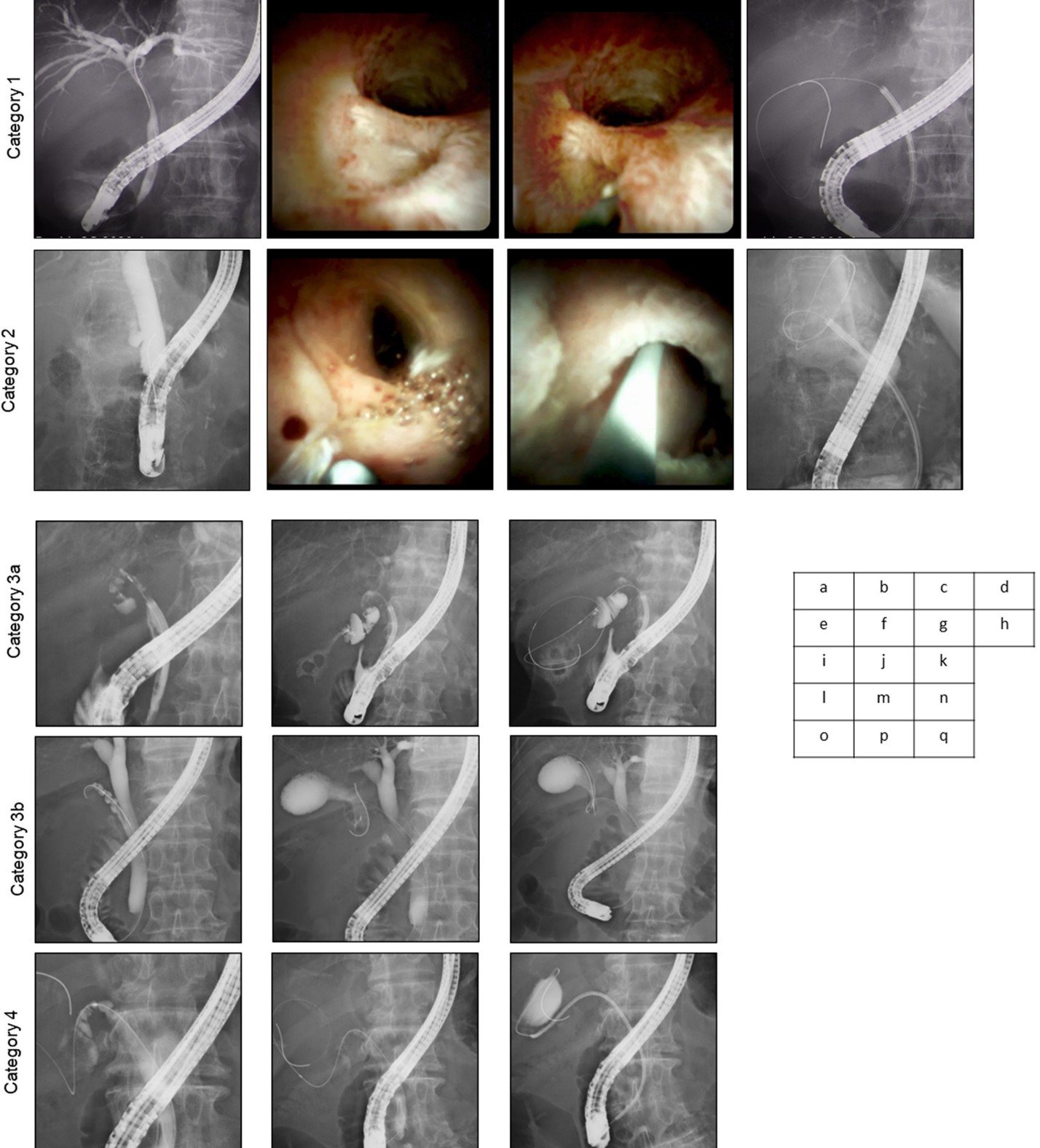

**Fig 1. Images showing the four-step classification system of ETGBD failure stage.** Fluoroscopic and endoscopic images show the features of each category. Category 1: a. failure to identify the cystic duct (CD) orifice. b, c, d. Pillar applied: cholangioscopic assistance. Category 2: e. failure to advance the GW across the CD takeoff due to unfavorable angle of guidewire (GW). f, g, h. Pillar applied: cholangioscopic assistance. Category 3a: i. failure of GW access to the gallbladder (GB) due to CD

obstruction (stone impaction). j, k. Pillar applied: 3-Fr microcatheter assistance. Category 3b: l. failure of GW access to the GB due to multiple tortuosities. m, n. Pillar applied: new flex-type GW. Category 4: o. failure of drainage stent insertion to the GB. p. Pillar applied: 3-Fr microcatheter assistance. q. Successful stent placement.

## Study design and outcomes

Patient characteristics, technical outcomes, procedure time (defined as time from starting seeking the CD to completion of stent deployment), and adverse events (AEs) of ETGBD were evaluated. To evaluate the efficacy of the optional Three-pillar assistance system, these were compared between the era of ETGBD with conventional techniques (Classical ETGBD) and that of innovative ETGBD using the four-step approach (Strategic ETGBD). In terms of the patient characteristics, age, sex, background characteristics as the reason for undergoing ETGBD, AC severity, presence of GB stones, inflammation, and condition of the papilla were evaluated. The primary outcome was the technical success rate of ETGBD, based on successful stent placement into the GB. Clinical success was defined as the resolution of clinical symptoms and laboratory findings associated with cholecystitis. AEs attributed to the performance of ETGBD were defined according to standard criteria, as follows. Pancreatitis was defined as the onset of new abdominal pain, with at least a three-fold elevation of serum amylase or lipase levels, at least 24 hours after the procedure [21]. Perforation was defined as retroperitoneal or bowel-wall perforation, as seen on any image. Hemorrhage was defined as clinical evidence of bleeding, with a decrease in hemoglobin $> 2$ g/dL or the need for endoscopic or transfusion treatment.

This retrospective study was approved by the Institutional Review Board (IRB) of Nagoya City University Hospital (approval No. 60-19-0219), in accordance with the Declaration of Helsinki ethical principles for medical research involving human participants. Written informed consent for ERCP was obtained from all patient. Because of the retrospective design of the study, the IRB waived the requirement for additional informed consent to have data from their medical records used in research. The study applied an "opt-out" option to obtain

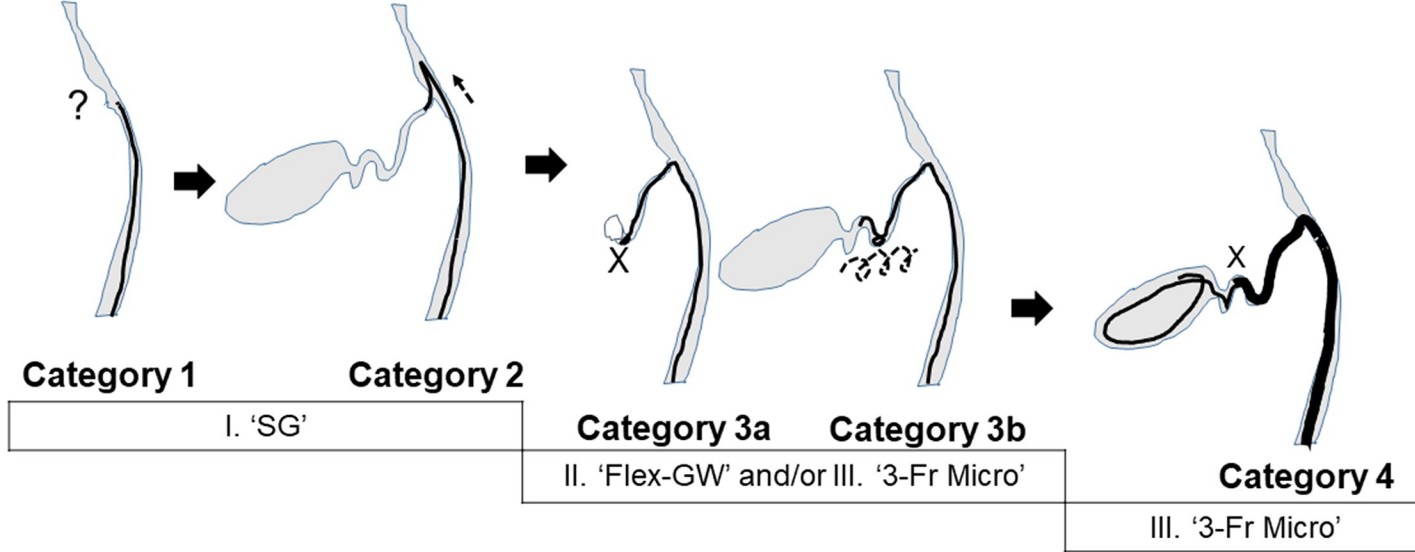

**Fig 2. Theoretical strategy of the four-step approach.** Diagrams show the pillars of assistance most useful for overcoming each potential failure category. Cholangioscopic assistance with SpyGlass DS (SG) can be helpful in troubleshooting to identify the CD orifice in Category 1 and to enable GW advancement across the CD takeoff in Category 2. The flex-type guidewire (Flex-GW) and 3-Fr microcatheter (3-Fr Micro) can be used in Category 3 to gain access to the GB in the case of CD obstruction (Category 3a) or multiple tortuosities (Category 3b). 3-Fr Micro can also be helpful in stent insertion to the GB in Category 4.

patient consent after being approved by the IRB of Nagoya City University Hospital. All data were fully anonymized before the data analysis.

## Statistical analysis

Data were analyzed using Fisher's exact probability test, the $m \times n$ test, or the Mann–Whitney U test, as appropriate. Values of $p<0.05$ were considered significant.

## Results

### Patient characteristics

This study enrolled 115 patients with AC (median age, 76 years; 74 male and 41 female) who were selected to undergo ETGBD. Of the overall patients, the severity of AC was mild in 65 (56.5%), moderate in 38 (33.0%), and severe in 12 (10.4%). GB stones were present in 88 (76.5%). Background characteristics as reason for ETGBD were as follows: comorbidity with common bile duct (CBD) stones (68 patients, 59.1%), antithrombotic therapy (22 patients, 19.1%), dementia with possible risk of self-removal of the PTGBD tube (20 patients, 17.4%), comorbidity with suspected GB cancer (16 patients, 13.9%), and the presence of ascites (13 patients, 11.3%) (Table 1). Among the 115 patients, the first 50 consecutive patients had undergone ETGBD with conventional techniques (Classical ETGBD group; 2008–2016), and the following 65 patients underwent ETGBD with the option of using the Three-pillar assistance system in the case of failure of conventional ETGBD (Strategic ETGBD group; 2017–2022). There were no significant differences in clinical characteristics between the two groups (Table 1).

### Application of Three-pillar assistance

Among the 65 patients in the Strategic ETGBD group, 37 (56.9%) underwent ETGBD without using any Three-pillar assistance. Based on the four-step approach, Flex-GW alone was applied

**Table 1. Patient characteristics.**

| | | | All ETGBD | | Classical ETGBD | | Strategic ETGBD | | P-value |
|---|---|---|---|---|---|---|---|---|---|
| | | | N = 115 | | N = 50 | | N = 65 | | |
| Age, y | median (range) | | 76 (40–96) | | 77 (40–94) | | 76 (43–96) | | 0.495 |
| Sex | (M:F) | | 74:41 | | 33:17 | | 41:24 | | 0.746 |
| Cholecystitis | | Mild | 65 | 56.5% | 31 | 62.0% | 34 | 52.3% | |
| | | Moderate | 38 | 33.0% | 12 | 24.0% | 26 | 40.0% | 0.154 |
| | | Severe | 12 | 10.4% | 7 | 14.0% | 5 | 7.7% | |
| WBC | mean±SD | (x10³/µL) | 11.3±5.44 | | 11.1±5.39 | | 11.8±5.47 | | 0.230 |
| CRP | mean±SD | (mg/dL) | 12.2±8.82 | | 12.6±8.72 | | 11.8±8.89 | | 0.647 |
| GB stone | | | 88 | 76.5% | 38 | 74.0% | 50 | 76.9% | 0.908 |
| Background | Comorbidity with CBD stone | | 68 | 59.1% | 32 | 64.0% | 36 | 55.4% | 0.352 |
| to challenge ETGBD | Anti-thrombotic agents | | 22 | 19.1% | 9 | 18.0% | 13 | 20.0% | 0.787 |
| | Dementia | | 20 | 17.4% | 10 | 20.0% | 10 | 15.4% | 0.267 |
| | Comorbidity with suspected GB cancer | | 16 | 13.9% | 9 | 18.0% | 7 | 10.8% | 0.517 |
| | Ascites | | 13 | 11.3% | 7 | 14.0% | 6 | 9.2% | 0.423 |
| Papilla | | Naïve | 78 | 67.8% | 34 | 68.0% | 44 | 67.7% | 0.972 |
| | | Post EST | 39 | 33.9% | 16 | 32.0% | 21 | 32.3% | |

CBD, common bile duct; CRP, C-reactive protein; ETGBD, endoscopic transpapillary gallbladder drainage; EST, endoscopic sphincterotomy; GB, gallbladder; WBC, white blood cell.

**Table 2. Application of the three pillars of assistance.**

|  | N = 65 |  |
|---|---|---|
| **Flex-GW** | | 19/65 | 29.2% |
| **SG** | | 15/65 | 23.1% |
| **3-Fr Micro** | | 11/65 | 16.9% |
| **None** | | 37/65 | 56.9% |
| Single-pillar | 15/65 | | |
| **SG alone** | | 7/65 | 10.8% |
| **Flex-GW alone** | | 7/65 | 10.8% |
| **3-Fr Micro alone** | | 1/65 | 1.5% |
| 2-pillars | 9/65 | | |
| **Flex-GW + 3-Fr Micro** | | 5/65 | 7.7% |
| **SG + Flex-GW** | | 3/65 | 4.6% |
| **SG + 3-Fr Micro** | | 1/65 | 1.5% |
| 3-pillars | 4/65 | | |
| **Flex-GW + SG + 3-Fr Micro** | | 4/65 | 6.2% |

Flex-GW, flex-type M-through guidewire; SG, SpyGlass DS; 3-Fr Micro, 3-Fr microcatheter.

most often, in 19 patients (29.2%), followed by SG alone in 15 patients (23.1%) and 3-Fr Micro alone in 11 patients (16.9%). Thirteen patients underwent ETGBD with various combined applications of the three pillars: Flex-GW+SG+3-Fr Micro, 4 patients (6.2%); Flex-GW+3-Fr Micro, 5 patients (7.7%); Flex-GW+SG, 3 patients (4.6%), and SG+3-Fr Micro, 1 patient (1.5%) (Table 2).

## Procedural outcomes

The success rates at each step category in the Classical ETGBD and Strategic ETGBD groups are shown in Fig 3A. In the combined groups, failure of ETGBD occurred most commonly at Category 1 (7 patients, 14.0%), followed by Category 3b (3 patients, 6.0%).

The pillars applied in the Strategic ETGBD group are shown in Fig 3B. Of patients in the Strategic ETGBD group, SG was applied following failure in 13/65 (20.0%) at Category 1 and in 2/65 (3.1%) at Category 2. Failure occurred at Category 3a in 2/65 (3.1%) and was averted by application of 3-Fr Micro. Failure occurred at Category 3b in 19/65 (29.2%), which was followed by application of both Flex-GW and 3-Fr Micro; however, the procedure was terminated in 2/19 patients. Failure occurred at Category 4 in 2/63 (3.2%) and was overcome by application of 3-Fr Micro. The correlation of the four-step classification to the severity grade of AC was shown in S1 Table.

The overall technical success rate was significantly higher in the Strategic ETGBD group (63/65 patients, 96.9%) compared with the Classical ETGBD group (36/50 patients; 72.0%) (p = 0.0001) (Fig 3A). The clinical success rate among patients in whom technical success was achieved was 91.7% (33/36) and 93.7% (59/63) in the Classical and Strategic ETGBD groups, respectively, and the difference lacked statistical significance (p = 0.711). The correlation of the severity grade of AC to the technical success rate was shown in S2 Table.

## Adverse Events (AEs)

CD injury was the most frequent AE in both groups (Classical ETGBD: 2/50, 4.0%; Strategic ETGBD: 5/65, 7.7%). In the Strategic ETGBD group, coordinated manipulation of Flex-GW and 3-Fr Micro following CD injury resulted in successful ETGBD in 4/5 cases. In contrast,

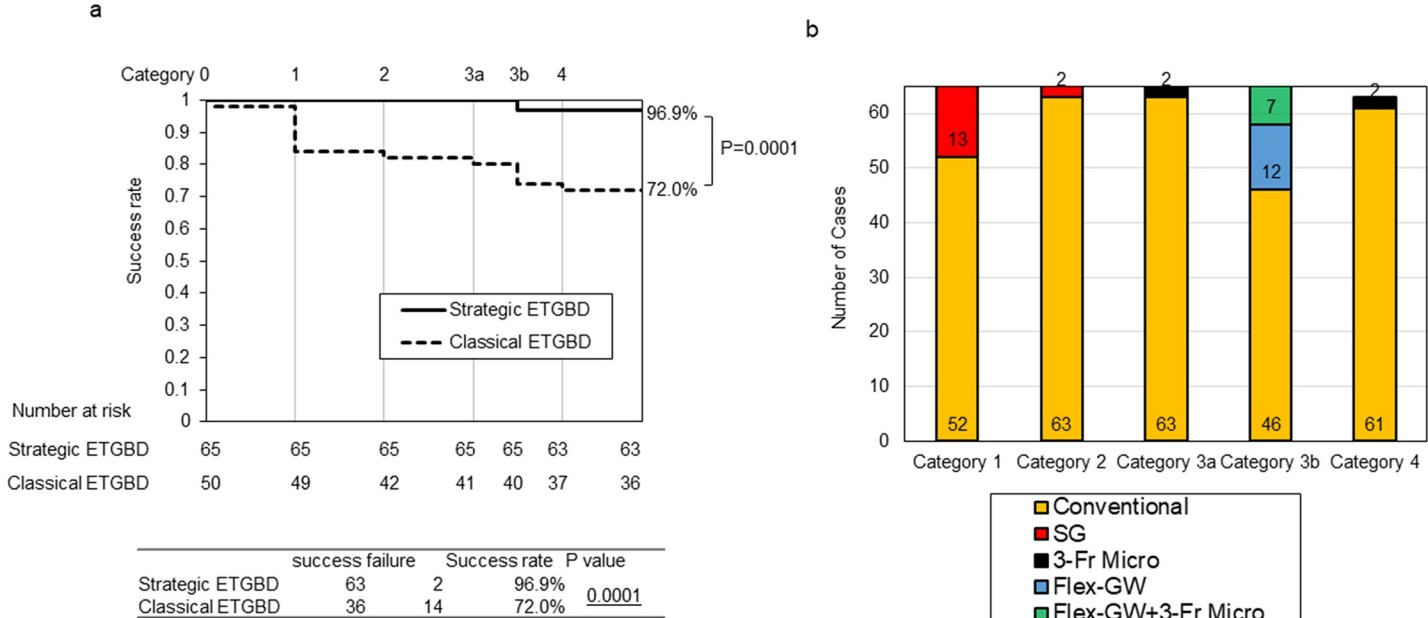

**Fig 3.** (a) Technical success of Classical ETGBD and Strategic ETGBD. Technical success is shown for each category. The overall success rate was significantly higher in the Strategic ETGBD group (63/65 patients, 96.9%) compared with the Classical ETGBD group (36/50 patients; 72.0%) (p = 0.0001). Solid line, Strategic ETGBD; broken line, Classical ETGBD. (b) Breakdown of techniques and pillars of assistance used at each category. The pillars of assistance applied at each category are shown. Of patients in the Strategic ETGBD group, SG was applied following failure in 13/65 (20.0%) at Category 1 and in 2/65 (3.1%) at Category 2. Failure occurred at Category 3a in 2/65 (3.1%) and was averted by application of 3-Fr Micro. Failure occurred at Category 3b in 19/65 (29.2%), which was followed by application of both Flex-GW and 3-Fr Micro. Failure occurred at Category 4 in 2/63 (3.2%) and was overcome by application of 3-Fr Micro. Conventional, ETGBD with conventional techniques; SG, cholangioscopic assistance; 3-Fr Micro, 3-Fr microcatheter assistance; Flex-GW, flex-type guidewire.

none of two could achieve ETGBD in the Classical ETGBD group. Other AEs were post-ERCP pancreatitis (Classical ETGBD: 2/50, 4.0%; Strategic ETGBD: 1/65, 1.5%), post-endoscopic sphincterotomy hemorrhage (Classical ETGBD: 1/50, 2.0%; Strategic ETGBD: 1/65, 1.5%), and peritoneal perforation (Classical ETGBD: 1/50, 2.0%) (Table 3). All patients that experienced AEs recovered following conservative observation.

## Procedure time

Overall Procedure time, which did not show significant differences between two groups, was shown in S3 Table. Procedure time defined as time from starting seeking the CD to completion of stent deployment was 17.5 min (range, 8–125 min) in the 36 successful patients in the

**Table 3. Complications.**

| | Conventional ETGBD | | ETGBD on Three pillars | |
| --- | --- | --- | --- | --- |
| | (N = 50) | | (N = 65) | |
| CD injury (GW penetration) | 2/50 | 4.0% | 5/65 | 7.7% |
| Post-ERCP pancreatitis | 2/50 | 4.0% | 1/65 | 1.5% |
| Post-EST hemorrhage | 1/50 | 2.0% | 1/65 | 1.5% |
| Peritoneal perforation | 1/50 | 2.0% | 0 | 0.0% |
| Total | 6/50 | 12.0% | 7/65 | 10.8% |

CD, cystic duct; ERCP, endoscopic retrograde cholangiopancreatography; EST, endoscopic sphincterotomy; ETGBD, endoscopic transpapillary gallbladder drainage; GW, guidewire.

**Table 4. Procedure time.**

| | Number | Procedure time (Median, min) | (range) | P-value |
|---|---|---|---|---|
| Successful Classical ETGBD | 36 | 17.5 | (8–125) | 0.90 |
| Successful Strategic ETGBD | 63 | 17 | (8–69) | |
| ↓ | | | | |
| ETGBD without any pillar | 37 | 12 | (8–37) | <0.0001 |
| ETGBD with one or more pillars | 26 | 31.5 | (9–69) | |

ETGBD, endoscopic transpapillary gallbladder drainage

Classical ETGBD group and 17 min (range, 8–69 min) in the 63 successful patients in the Strategic ETGBD group, with no significant difference (Table 4).

Of the successful cases in the Strategic ETGBD group, 26/63 underwent successful ETGBD with one or more pillars and 37/63 underwent successful ETGBD with conventional approaches without any pillars. Unsurprisingly, in the Strategic ETGBD group, time to complete the procedure was significantly longer in those with one or more pillars compared with those without any pillars (median time [range]: 31.5 min [9–69 min] vs. 12 min [8–37 min], respectively; p<0.0001).

## Discussion

Early cholecystectomy has become the definitive treatment for AC [3, 4], but in cases of patients who are unfit for surgery, decompressive drainage may be required as a life-saving procedure. Currently, three access routes; 1) PTGBD, 2) endoscopic ultrasound-guided GB drainage (EUS-GBD), and 3) ETGBD, are available to drain the GB in AC as alternative to surgery. PTGBD is traditionally the first-line approach because of its minimally invasive technique with a low complication [6–8]. However, percutaneous drainage cannot be performed in patients with massive ascites, an anatomically inaccessible GB, those at risk of self-removal of the drainage tube, or with a bleeding tendency [9]. EUS-GBD is recognized as another alternative approach [15, 22–24]. Transmural GB drainage by EUS has generated an interest in the feasibility and effectiveness, although it should be performed in high volume tertiary care centers by skillful endoscopists. However, this technique is also contraindicated for patients with serious coagulopathy, thrombocytopenia, or ascites. ETGBD has been reported as a novel alternative GB drainage procedure for patients with AC [11–15]. ETGBD can be performed in the same ERCP session to treat CBD stones. ETGBD can preserve the normal GB structure without the need to create a fistula, which brings a suitable indication for patients with serious coagulopathy, thrombocytopenia, or ascites. In our practical applications during this period, fundamental strategies for the patients of AC in whom are unfit for early surgery (N = 380) are as follows:1) PTGBD is the first-line drainage procedure (N = 219). 2) When PTGBD cannot be performed in patients with an anatomically inaccessible GB, those at risk of self-removal of the drainage tube, EUS-GBD (N = 46) or ETGBD (N = 115) is the second-line drainage procedure. 3) In patients with possible CBD stones, suspected GB cancer, serious coagulopathy and thrombocytopenia, or ascites, ETGBD is preferentially performed as the alternative technique, as shown in S1 Fig and Table 1.

ETGBD has not yet been established as a standard procedure because it requires advanced endoscopic techniques. A recent meta-analysis reported a technical success rate of 83% for ETGBD [15]. TG2018 guidelines note that ETGBD should be undertaken only by skilled pancreatobiliary endoscopists at high-volume institutes [5]. Endoscopic ultrasound-guided GB drainage has been reported as an alternative GB drainage procedure [15, 22–24]. However, in

particular situations such as pre-surgery malignancy, massive ascites, or other high-risk comorbid conditions with anti-thrombotic agents, the endoscopic transpapillary approach remains the only choice to achieve GB drainage. A theoretical ETGBD strategy that overcomes the current limitations is clearly desirable.

Several case reports have reported the usefulness of cholangioscopy-assisted technique as an emerging solution [25–28]. Ridtitid et al also reported its usefulness in a retrospective study of 104 patients with AC [29]. However, they also stated that, compared with CD cannulation, cholangioscopic assistance has a limited contribution for gallbladder GW placement and GB stent insertion. In fact, additional cholangioscopic assistance increased the technical success rate from 53% to 75% according to the study.

We have previously reported that ETGBD procedures can be classified into a series of steps, and that a four-step classification system is helpful for appreciating issues that can complicate ETGBD [16]. When the four-step system is employed, cholangioscopic assistance can function for troubleshooting at Category 1, failure to identify the CD orifice, and Category 2, failure to advance the GW across the CD takeoff due to unfavorable angle, but not for the subsequent steps. Category 3, failure of GW access to the GB due to CD obstruction (Category 3a) or to multiple tortuosities (Category 3b) are also major issues that must be overcome to improve the limited success rate of ETGBD. Accordingly, we adopted two additional techniques in the Three-pillar assistance system. Flex-GW has been designed with an innovative flexible tip portion that allows smooth tracking and easy maneuverability [17], and is expected to be particularly valuable for advancing through a tortuous cystic duct. 3-Fr Micro was designed to overcome various limitations of the ERCP procedure [19]. The major benefits of this microcatheter stem from its flexibility and slimness. Contrast medium can be injected through the microcatheter and it can also accept a 0.025-inch GW for over-the-wire manipulation, which can assist in GW maneuverability, as suggested by its established use in selective angiography.

Coordinated manipulation with Flex-GW and 3-Fr Micro can achieve the desired effect of passage through a bothersome CD. Moreover, even after successful GW advancement to the GB, there is the possibility of failure of drainage stent insertion to the GB, as in Category 4. The 3-Fr Micro demonstrates an extraordinary ability to approach the GB. When a conventional ERCP catheter could not be advanced through a CD that was deeply crooked and/or severely stenosed, the coaxial microcatheter was able to follow the GW over the flexure, allowing successful contrast injection through the microcatheter to confirm the cavity of the GB, and bile juice in the GB was aspirated via the microcatheter. Decompression of the GB and subsequent GW exchange with a stiffer type loosened the crooked CD, which enabled a plastic stent to be deployed.

Based on their independent advantages, we considered these three devices as pillars of stability and incorporated them in a newly designed theoretical strategy for ETGBD. Employing this theoretical strategy in the present study resulted in a technical success rate of 96.9%, which was significantly higher than that of the conventional approach.

The three pillar devices used in the system can be used independently for troubleshooting, and also in a coordinated way. There is no catheter for cannulation in cholangioscopy because of the narrow working channel; however, a 3-Fr Micro could be applied as a custom-made catheter for cholangioscopy, for injection of contrast medium and in assistance with GW manipulation. The combinations of SG plus Flex-GW, SG plus 3-Fr Micro, Flex-GW plus 3-Fr Micro, or all three pillars could facilitate GW advancement to the GB in patients in whom multiple-step issues exist.

There was no significant difference between the earlier and later ETGBD periods in terms of procedure time to successful GB stenting; however, when one or more pillars were applied, significantly more time was required to complete the procedure compared with conventional

ETGBD, which is to be expected. The median procedure times of 17.5 min and 12 min for successful Classical ETGBD and for ETGBD without any pillars in Strategic ETGBD, respectively, might indicate the amount of experimental time required for successful ETGBD with the conventional approach, and might serve as a useful reference regarding the timing that should be allowed for the Three-pillar system. CD injury due to GW penetration was the most frequent AE, in Strategic ETGBD (5/65, 7.7%), but strategic ETGBD did not increase the total incidence of AEs (Table 3). It has been reported that CD injury reduced the technical success rate of ETGBD [30]. In the present study, however, the coordinated manipulation of Flex-GW and 3-Fr Micro 4 resulted in successful ETGBD in 4/5 cases. In contrast, CD injury was observed in only 2 cases of Classical ETGBD (4.0%), but ETGBD could not be achieved in either of these. These devices might also be promising troubleshooters for cases of CD perforation during ETGBD.

The appropriate use of the three-pillar system should also be considered from the perspective of cost-effectiveness: SG, USD 2,039 (JPY 300,000); Flex-GW, USD 156 (JPY 23,000); 3-Fr Micro, USD 169 (JPY 25,000) (currency exchange rate, USD/JPY: 0.0068/1). This strategy could be used along with four-step classification to help identify the steps causing potential problems and assist in appropriate device application.

Limitations of this retrospective study are its small sample size and that it was a single-center study. Further studies with larger numbers in multicenter prospective settings are needed.

In conclusion, a theoretical Step strategy that employed Three-pillar assistance achieved significantly higher rates of successful ETGBD compared with the conventional technique. Coordination of the pillars of SG, Flex-GW, and 3-Fr Micro with a four-step classification approach for identifying complicating issues appears useful for improving the success rate of ETGBD.

## Supporting information

**S1 Appendix. Lists of the patients recruited in this study compared to the previous publication.**
(DOCX)

**S1 Table. The correlation of the four-step classification to the severity of grade of AC.**
(DOCX)

**S2 Table. The correlation of the severity of grade of AC to the technical success rate.**
(DOCX)

**S3 Table. Overall procedure time.**
(DOCX)

**S1 Fig. The flow chart of the management of AC.** In our practical applications during this period, fundamental strategies for the patients of acute cholecystitis (AC) in whom are unfit for early surgery (N = 380) are as follows:1) Percutaneous transhepatic gallbladder drainage (PTGBD) is the first-line drainage procedure (N = 219). 2) When PTGBD cannot be performed in patients with an anatomically inaccessible GB, those at risk of self-removal of the drainage tube, endoscopic ultrasound-guided GB drainage (EUS-GBD) (N = 46) or endoscopic transpapillary GB drainage (ETGBD) (N = 115) is the second-line drainage procedure. 3) In patients with possible common bile duct (CBD) stones, suspected GB cancer, serious coagulopathy and thrombocytopenia, or ascites, ETGBD is preferentially performed as the alternative technique.
(TIF)

## Author Contributions

**Conceptualization:** Michihiro Yoshida, Itaru Naitoh.

**Data curation:** Yasuki Hori, Kenta Kachi, Go Asano, Hidenori Sahashi.

**Formal analysis:** Akihisa Kato.

**Project administration:** Kazuki Hayashi.

**Supervision:** Hiromi Kataoka.

**Validation:** Tadashi Toyohara, Kayoko Kuno, Yusuke Kito.

**Writing – original draft:** Michihiro Yoshida.

**Writing – review & editing:** Itaru Naitoh.

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
