## [Decision Letter · Decision Letter 0]

18 Oct 2022

PONE-D-22-21361Theoretical ‘Step’ Approach with ‘Three-pillar’ Device Assistance for Successful Endoscopic Transpapillary Gallbladder DrainagePLOS ONE

Dear Dr. Naitoh,

Thank you for submitting your manuscript to PLOS ONE. After careful consideration, we feel that it has merit but does not fully meet PLOS ONE’s publication criteria as it currently stands. Therefore, we invite you to submit a revised version of the manuscript that addresses the points raised during the review process.

We look forward to receiving your revised manuscript.

Kind regards,

Alejandro Piscoya

Academic Editor

PLOS ONE

Journal Requirements:

" NO. The funders had no role in study design, data collection and analysis, decision to publish, or preparation of the manuscript."

Reviewers' comments:

Reviewer's Responses to Questions

**Comments to the Author**

1. Is the manuscript technically sound, and do the data support the conclusions?

Reviewer #1: Yes

Reviewer #2: Yes

2. Has the statistical analysis been performed appropriately and rigorously? 

Reviewer #1: Yes

Reviewer #2: Yes

3. Have the authors made all data underlying the findings in their manuscript fully available?

Reviewer #1: Yes

Reviewer #2: No

4. Is the manuscript presented in an intelligible fashion and written in standard English?

Reviewer #1: Yes

Reviewer #2: Yes

5. Review Comments to the Author

Reviewer #1: Many thanks for giving me the opportunity to review this manuscript.

In the submitted study the authors evaluated the clinical role of endoscopic transpapillary gallbladder drainage in patients with acute cholecystitis.

In this regard four clinical scenarios were defined that might impair technical access to the gallbladder and therefore proper drainage.

A total of 115 patients was retrospectively included evaluating conventional ETGBD technique

(Classical ETGBD, N=50) and strategic ETGBD with optional Three-pillar assistance

(Strategic ETGBD, N=65).

The overall technical success rate was significantly higher for the strategic approach 96.9% compared to the classical approach.

In addition, technical methods were evaluated for each clinical scenario in the strategic approach.

Overall assessment:

The study is of great interest for the endoscopic treatment of patients with acute cholecystitis. The manuscript is technical sound, the data properly support the stated conclusion.

Nevertheless, some major points should be addressed:

Methods:

- The authors already published some studies referring to the same classification system and technical options.

I assume that there is a significant overlap of the patients recruited in this retrospective study compared to the enumerated publications (reference numbers 16 - 19). The potential overlap of patients needs to be clearly listed/defined.

- The word "step" for the different clinical scenarios/situations is confusing to my opinion. I would prefer to scenario or category 0-4 for the different "steps".

Results:

- An additional correlation of the "steps" to the severity grade of acute cholecystitis and to the respective technical success rates would be helpful to elucidate potential bias that is caused by the inflammation itself.

- The authors only refer to the technical success rates. What about the clinical success rates? Did the patients benefit from the endoscopic interventions? That should be addressed additionally.

- Figure 3b: In the legend the pictograms of "conventional", "SG" and "Flex GW" cannot be distinguished (at least in the pdf that was send)

Discussion:

Technically, three access routes are available to drain the gallbladder in acute cholecystitis as alternative to surgery:

(i) Transcutaneous GB-drainage

(ii) transluminal GB-drainage (cholecysto-duodenostomy or cholecysto-gastrostomy with LAMS)

(iii) transpapillary drainage

The authors should discuss the other options in more detail. Especially, what are the pros and cons when comparing the transluminal to the transpapillary approach. Please suggest a respective patient selection as a clinical guidance.

Reviewer #2: This study is evaluating a theoretical step strategy for the ETGBD. The authors concluded that it would improve the technical success rate of ETGBD. It was well written, but there were several points to be clarified.

1. In mild acute cholecystitis, it is not usually necessary to do GB drainage. All the patients with mild cholecystitis underwent GB drainage? What is the strategy for acute cholecystitis in your institution? During the study period, PTGBD or EUS-GBD was not performed at all?

2. In the comparison of procedure time, you compare the procedure time defined as time from starting seeking the CD to completion of stent deployment. As the novel approach requires SG insertion before starting the seeking the CD, it was not fair to compare the procedure time under defined conditions. Please provide the overall procedure time for both groups.

3. In table 2, it looks procedure time was shorter in success strategic ETGBD without any pillar than in success conventional ETGBD. What is the main reason for this difference?

4. When did you decide to use each pillar in each step. For example. how long did you continue to seek CD without SG?

5. Please provide the cost of each procedure. It is very important point to select new strategic ETGBD.

6. PLOS authors have the option to publish the peer review history of their article (what does this mean?). If published, this will include your full peer review and any attached files.

Reviewer #1: **Yes: **Prof. Dr. Mark Ellrichmann

Reviewer #2: No

---

## [Author Response · Author response to Decision Letter 0]

5 Nov 2022

We are pleased that you are interested in our paper, and reviewer’s comments are extremely helpful for our manuscript to become a more substantial paper.

We have answered all reviewer’s comments adequately. We highlighted the changes to the revised manuscript by colored text ‘Revised Manuscript with Track Changes’. A point-by-point reply to the Journal Requirements and the reviewer’s comments is as follows.

Journal Requirements:

We ensured PLOS ONE’S style requirements and modified file naming.

2. Thank you for stating the following financial disclosure:　"NO. The funders had no role in study design, data collection and analysis, decision to publish, or preparation of the manuscript."　At this time, please address the following queries:　d) If you did not receive any funding for this study, please state: “The authors received no specific funding for this work.”

We did not receive any funding for this study. We would like to change this statement in the online submission form as follows: The authors received no specific funding for this study.

Review Comments to the Author

Reviewer #1

Methods:

- The authors already published some studies referring to the same classification system and technical options. I assume that there is a significant overlap of the patients recruited in this retrospective study compared to the enumerated publications (reference numbers 16 - 19). The potential overlap of patients needs to be clearly listed/defined.

Thank you for the suggestion. We added a list of patients recruited in this study, as S1 Appendix, to clarify the overlap of patients compared to the enumerated publications (Ref.16-19).

- The word "step" for the different clinical scenarios/situations is confusing to my opinion. I would prefer to scenario or category 0-4 for the different "steps".

As suggested, we reworded ‘step’ for the different clinical situations to ‘Category 0-4’. Thank you for the suggestion.

Results:

- An additional correlation of the "steps" to the severity grade of acute cholecystitis and to the respective technical success rates would be helpful to elucidate potential bias that is caused by the inflammation itself.

We added the correlation of the ‘steps (Category)’ to the severity grade of acute cholecystitis and to the respective technical success rates in S1 and S2 Tables. I appreciate your pointing.

- The authors only refer to the technical success rates. What about the clinical success rates? Did the patients benefit from the endoscopic interventions? That should be addressed additionally.

The clinical success rate among patients in whom technical success was achieved was 91.7% (33/36) and 93.7% (59/63) in the Classical and Strategic ETGBD groups, respectively, and the difference lacked statistical significance (p=0.711). We added the clinical success rates in the ‘Procedural outcomes’ part (Results part).

- Figure 3b: In the legend the pictograms of "conventional", "SG" and "Flex GW" cannot be distinguished (at least in the pdf that was send)

Thank you for pointing. We revised them for the clear-cut pictograms.

Discussion:

Technically, three access routes are available to drain the gallbladder in acute cholecystitis as alternative to surgery: (i) Transcutaneous GB-drainage (ii) transluminal GB-drainage (cholecysto-duodenostomy or cholecysto-gastrostomy with LAMS) (iii) transpapillary drainage. The authors should discuss the other options in more detail. Especially, what are the pros and cons when comparing the transluminal to the transpapillary approach. Please suggest a respective patient selection as a clinical guidance.

We agree with this point. As suggested, we added the discussion about the other options in detail, including the pros and cons in the top of the Discussion part. Thank you for the suggestion.

Reviewer #2: 

1. In mild acute cholecystitis, it is not usually necessary to do GB drainage. All the patients with mild cholecystitis underwent GB drainage? What is the strategy for acute cholecystitis in your institution? During the study period, PTGBD or EUS-GBD was not performed at all?

Thank you for pointing. In our practical applications, fundamental strategies for the patients of AC in whom are unfit for early surgery are as follows:1) PTGBD is the first-line drainage procedure. 2) When PTGBD cannot be performed in patients with an anatomically inaccessible GB, those at risk of self-removal of the drainage tube, EUS-GBD or ETGBD is performed as the second-line drainage procedure. 3) In patients with possible CBD stones, suspected GB cancer, serious coagulopathy and thrombocytopenia, or ascites, ETGBD is preferentially performed. We added these explanations of the strategy in the Discussion part.

2. In the comparison of procedure time, you compare the procedure time defined as time from starting seeking the CD to completion of stent deployment. As the novel approach requires SG insertion before starting the seeking the CD, it was not fair to compare the procedure time under defined conditions. Please provide the overall procedure time for both groups.

You raised an important point. We agree the importance of the overall procedure time. But, because of the nature of the retrospective study, the overall procedure time involved many participation factors such as the difficulty of biliary cannulation, the presence and extent of CBD stones requiring the extraction, and with or without the necessity of pathological examination, which must have a huge effect on the overall procedure time. So, we defined the procedure time as time from starting seeking the CD to completion of stent deployment to exclude these possible influencing factors. We will leave the analysis of the overall procedure time to future prospective studies. Thank you for pointing.

3. In table 2, it looks procedure time was shorter in success strategic ETGBD without any pillar than in success conventional ETGBD. What is the main reason for this difference?

Thanks for your good insight. When conventional ETGBD was performed, we did not have any effective troubleshooter such as three-pillar Device assistance. In difficult cases of conventional ETGBD, it took a lot of work and time to continue to try ETGBD without quitting. In contrast, strategic ETGBD without any pillar tended to involve easy cases.

4. When did you decide to use each pillar in each step. For example. how long did you continue to seek CD without SG?

We empirically drew on the procedure time of successful Classical ETGBD (median, 17.5 min). Although it depended on the endoscopist’s preference because of the nature of the retrospective study, we basically continued to seek CD without SG for 15-20 min.

5. Please provide the cost of each procedure. It is very important point to select new strategic ETGBD.

The cost of each procedure is as follows: SG, USD 2,039 (JPY 300,000); Flex-GW, USD 156 (JPY 23,000); 3-Fr Micro, USD 169 (JPY 25,000) (currency exchange rate, USD/JPY: 0.0068/1). I added this information in the Discussion part.

Thank you very much for your consideration of our paper. We look forward to hearing from you again.

Sincerely yours,

Itaru Naitoh

---

## [Decision Letter · Decision Letter 1]

22 Dec 2022

PONE-D-22-21361R1Theoretical ‘Step’ Approach with ‘Three-pillar’ Device Assistance for Successful Endoscopic Transpapillary Gallbladder DrainagePLOS ONE

Dear Dr. Naitoh,

Thank you for submitting your manuscript to PLOS ONE. After careful consideration, we feel that it has merit but does not fully meet PLOS ONE’s publication criteria as it currently stands. Therefore, we invite you to submit a revised version of the manuscript that addresses the points raised during the review process.

We look forward to receiving your revised manuscript.

Kind regards,

Alejandro Piscoya

Academic Editor

PLOS ONE

Journal Requirements:

Reviewers' comments:

Reviewer's Responses to Questions

**Comments to the Author**

1. If the authors have adequately addressed your comments raised in a previous round of review and you feel that this manuscript is now acceptable for publication, you may indicate that here to bypass the “Comments to the Author” section, enter your conflict of interest statement in the “Confidential to Editor” section, and submit your "Accept" recommendation.

Reviewer #1: All comments have been addressed

Reviewer #2: (No Response)

2. Is the manuscript technically sound, and do the data support the conclusions?

Reviewer #1: Yes

Reviewer #2: Yes

3. Has the statistical analysis been performed appropriately and rigorously? 

Reviewer #1: Yes

Reviewer #2: Yes

4. Have the authors made all data underlying the findings in their manuscript fully available?

Reviewer #1: Yes

Reviewer #2: No

5. Is the manuscript presented in an intelligible fashion and written in standard English?

Reviewer #1: Yes

Reviewer #2: Yes

6. Review Comments to the Author

Reviewer #1: The authors`sufficiently answered alls question that were raised, the manuscript was adapted accordingly.

Many thanks for the opportunity to review this manuscript.

Reviewer #2: It was well revised and written, but there were some point to be clarified.

1. Though this study is a retrospective nature, it would be important to show overall procedure time in both groups.

2. Please provide overall procedure time in failed cases in both groups to minimize the potential selection biases.

3. In the study periods, how many patients underwent gallbladder drainage other than ETGBD. Please provide the flow chart of the management of acute cholecystitis during this periods.

7. PLOS authors have the option to publish the peer review history of their article (what does this mean?). If published, this will include your full peer review and any attached files.

Reviewer #1: **Yes: **Prof. Dr. Mark Ellrichmann

Reviewer #2: No

---

## [Author Response · Author response to Decision Letter 1]

23 Dec 2022

We are pleased that you are interested in our paper, and reviewer’s comments are helpful for our manuscript to become a more substantial paper.

We have answered all reviewer’s comments adequately. We highlighted the changes to the revised manuscript by colored text ‘Revised Manuscript with Track Changes R2’. A point-by-point reply to the reviewer’s comments is as follows.

Reviewer #2: It was well revised and written, but there were some point to be clarified.

1. Though this study is a retrospective nature, it would be important to show overall procedure time in both groups.

As suggested, we added overall procedure time of both groups in the ‘Procedure time’ part (Results part) and S3 table. Thank you for the suggestion.

2. Please provide overall procedure time in failed cases in both groups to minimize the potential selection biases.

As suggested, we also added overall procedure time of failed cases in the ‘Procedure time’ part (Results part) and S3 table. Thank you for the suggestion.

3. In the study periods, how many patients underwent gallbladder drainage other than ETGBD. Please provide the flow chart of the management of acute cholecystitis during this periods.

In our practical applications during this period, fundamental strategies for the patients of AC in whom are unfit for early surgery (N=380) are as follows:1) PTGBD is the first-line drainage procedure (N=219). 2) When PTGBD cannot be performed in patients with an anatomically inaccessible GB, those at risk of self-removal of the drainage tube, EUS-GBD (N=46) or ETGBD (N=115) is performed as the second-line drainage procedure. 3) In patients with possible CBD stones, suspected GB cancer, serious coagulopathy and thrombocytopenia, or ascites, ETGBD is preferentially performed. 

We updated these explanations of the strategy in the Discussion part and added the flow chart of the management of AC during this period in S1 Figure.

Thank you for the suggestion.

---

## [Editor Report · Decision Letter 2]

27 Jan 2023

Theoretical ‘Step’ Approach with ‘Three-pillar’ Device Assistance for Successful Endoscopic Transpapillary Gallbladder Drainage

PONE-D-22-21361R2

Dear Dr. Naitoh,

We’re pleased to inform you that your manuscript has been judged scientifically suitable for publication and will be formally accepted for publication once it meets all outstanding technical requirements.

Kind regards,

Alejandro Piscoya

Academic Editor

PLOS ONE
---

## [Editor Report · Acceptance letter]

1 Feb 2023

PONE-D-22-21361R2 

Theoretical Step Approach with ‘Three-pillar’ Device Assistance for Successful Endoscopic Transpapillary Gallbladder Drainage 

Dear Dr. Naitoh:

I'm pleased to inform you that your manuscript has been deemed suitable for publication in PLOS ONE. Congratulations! Your manuscript is now with our production department. 

Kind regards, 

on behalf of

Professor Alejandro Piscoya 

Academic Editor

PLOS ONE